# Runx2-Twist1 interaction coordinates cranial neural crest guidance of soft palate myogenesis

Xia Han[1], Jifan Feng[1], Tingwei Guo[1], Yong-Hwee Eddie Loh[2], Yuan Yuan[1], Thach-Vu Ho[1], Courtney Kyeong Cho[1], Jingyuan Li[1], Junjun Jing[1], Eva Janeckova[1], Jinzhi He[1], Fei Pei[1], Jing Bi[1], Brian Song[1], Yang Chai[1]*

[1]Center for Craniofacial Molecular Biology, University of Southern California, Los Angeles, Los Angeles, United States; [2]USC Libraries Bioinformatics Services, University of Southern California, Los Angeles, Los Angeles, United States

**Abstract** Cranial neural crest (CNC) cells give rise to bone, cartilage, tendons, and ligaments of the vertebrate craniofacial musculoskeletal complex, as well as regulate mesoderm-derived craniofacial muscle development through cell-cell interactions. Using the mouse soft palate as a model, we performed an unbiased single-cell RNA-seq analysis to investigate the heterogeneity and lineage commitment of CNC derivatives during craniofacial muscle development. We show that Runx2, a known osteogenic regulator, is expressed in the CNC-derived perimysial and progenitor populations. Loss of *Runx2* in CNC-derivatives results in reduced expression of perimysial markers (*Aldh1a2* and *Hic1*) as well as soft palate muscle defects in *Osr2-Cre;Runx2^{fl/fl}* mice. We further reveal that Runx2 maintains perimysial marker expression through suppressing *Twist1,* and that myogenesis is restored in *Osr2-Cre;Runx2^{fl/fl};Twist1^{fl/+}* mice. Collectively, our findings highlight the roles of Runx2, Twist1, and their interaction in regulating the fate of CNC-derived cells as they guide craniofacial muscle development through cell-cell interactions.

*For correspondence:
ychai@usc.edu

Competing interests: The authors declare that no competing interests exist.

## Introduction

The craniofacial musculoskeletal complex is an important evolutionary innovation in vertebrates that facilitates feeding, breathing, facial expression, and verbal communication. One unique component of this complex is the cranial neural crest (CNC) cells. CNC cells and their derivatives give rise to all facial bones, ligaments, and muscle connective tissues including tendons and fascia that directly surround muscle cells (*Chai and Maxson, 2006*; *Heude et al., 2010*; *Le Douarin et al., 2004*). Recently, CNC cells have been shown to regulate formation of mesoderm-derived craniofacial muscles through cell-cell interactions. Mouse genetic studies have further shown that CNC cells and their derivatives surround myogenic cells, facilitate myogenic cell migration, and establish cellular scaffolding at future myogenic sites to regulate muscle morphogenesis (*Grimaldi et al., 2015*; *Han et al., 2014*; *Rinon et al., 2007*). For instance, disruption of *Dlx5/6*, which is specifically expressed by CNC-derived cells in the mouse, leads to the loss of all first pharyngeal arch-derived masticatory muscles and second pharyngeal arch-derived muscles (*Heude et al., 2010*). Proliferation and survival of CNC-derived cells and fourth to sixth pharyngeal arch-derived myogenic cells in the soft palate are also affected, resulting in a truncated soft palate in *Dlx5^{-/-}* mice (*Sugii et al., 2017*). Similarly, TGFβ signaling in CNC-derived cells is critical for proliferation and differentiation of tongue and masseter muscle cells (*Han et al., 2014*; *Hosokawa et al., 2010*; *Iwata et al., 2013*). It is important to note that the transcription factors and signaling pathways critical for the role of CNC-derived cells in myogenesis are not restricted in their expression to merely the CNC-derived cells surrounding the muscle, known as perimysial cells; they are also expressed in other CNC-derived

musculoskeletal tissues (e.g. bones, bone eminences, and tendons) and regulate their development (*Depew et al., 2002*; *Hosokawa et al., 2010*; *Zhao et al., 2008*). This suggests that the same transcription factors and signaling pathways could activate cell-type-specific responses in multiple components of the musculoskeletal complex that may help coordinate the development of this intricate system. Therefore, it is important to investigate the cell-type-specific signaling mechanisms that regulate the heterogeneous CNC-derived cells and reveal their impact on craniofacial musculoskeletal development.

The soft palate is a muscular structure that comprises the posterior third of the palate. Its movement opens and closes the nasopharynx and oral cavity to direct air and food into different passages, as well as during speech. Several components of the soft palate are CNC-derived, including perimysial cells, palatal stromal cells that constitute the majority of palatal shelf mesenchyme, and tendons. In contrast, the soft palatal muscles are derived from pharyngeal mesoderm (*Grimaldi et al., 2015*). Five muscles are involved in the human soft palate. They include the tensor veli palatini (TVP) and levator veli palatini (LVP), which descend from the skull base and elevate the soft palate, and the palatoglossus (PLG) and palatopharyngeus (PLP), which ascend from the tongue and the pharyngeal wall, respectively, and depress the soft palate (*Li et al., 2019*). The fifth muscle, the musculus uvulae, which is specific to humans, is located at the end of the soft palate. Patients with cleft palate often have multiple types of tissue abnormalities including bone defects and insufficient, misoriented muscle fibers (*Dixon et al., 2011*; *Li et al., 2019*). Functional restoration of cleft soft palate is challenging because the muscles have limited ability to regenerate after surgical repair of the cleft (*Von den Hoff et al., 2019*). Therefore, comprehensive understanding of the growth and transcription factors that regulate the coordinated development of the distinct tissues in the soft palate is of both scientific and clinical significance.

Runx2, a known regulator of skeletogenesis and odontogenesis, is a Runt DNA-binding domain family transcription factor and contains multiple activation and repression domains. Patients with haploinsufficiency of *RUNX2* exhibit cleidocranial dysplasia, which is associated with specific skeletal and dental phenotypes. During osteoblast differentiation, Runx2 acts as a master organizer, recruiting phosphorylated Smad1/5, c-Fos, and c-Jun to activate expression of osteoblast-specific collagen and fibronectin upon receiving BMP signals and parathyroid hormones; it also binds histone deacetylases to repress cell cycle inhibitors and stimulate proliferation (*Schroeder et al., 2005*). Despite its well-known roles in regulating hard tissue development, the importance of Runx2 in soft tissue development has not been studied. Interestingly, several clinical case reports reveal that some RUNX2-deficient patients have thin masseter muscles, cleft lip, or high-arched palate (*Furuuchi et al., 2005*; *Sapp et al., 2004*; *Sull et al., 2008*; *Yamachika et al., 2001*). These studies hint that Runx2 may regulate the development of the palatal muscles and other components in sync with the bone to form the intricate craniofacial musculoskeletal complex by performing multiple tissue-specific roles.

In this study, we performed an unbiased transcriptional profile analysis of the developing soft palate using single-cell RNA-seq (scRNA-seq). We identified cellular-level heterogeneity in the CNC-derived soft palate mesenchyme, associated with distinctive cell fates: perimysial and midline mesenchymal lineages, as well as previously unknown cell types associated with putative progenitors. In addition, we found Runx2 was expressed in non-osteochondrogenic cells in the perimysial populations and in CNC-derived progenitor cells. Consistent with its expression pattern, loss of *Runx2* in CNC-derived cells resulted in a soft palate cleft along with tendon, bone, and muscle differentiation defects. We further revealed that loss of *Runx2* led to ectopic expression of *Twist1* and reduction in the expression of perimysial marker genes (*Aldh1a2* and *Hic1*) in CNC-derived perimysial cells. We also identified that suppression of *Twist1* expression by Runx2 is important for the development of palatal muscles and for maintaining the expression of the perimysial marker and myogenic-promoting gene *Aldh1a2*, thus coordinating soft palate morphogenesis by orchestrating the fate determination of CNC-derived mesenchymal lineages. Taken together, our findings reveal that Runx2 regulates distinct downstream targets in different subgroups of CNC-derived cells to fine-tune the development of craniofacial structures.

## Results

### Single-cell RNA-seq analysis reveals mesenchymal cell heterogeneity within the soft palate primordium

CNC-derived cells adopt diverse fates to establish the soft palate during development. To investigate the heterogeneity of the CNC-derived population that contributes to the developing soft palate at the single-cell level, we performed unbiased single-cell RNA-seq and integration analysis at three critical stages (E13.5, E14.5, E15.5). The soft palate primordium begins to form around E13.5, followed by fusion of the soft palatal shelves at E14.5 and myotube maturation at E15.5 (*Li et al., 2019*). Following integration analysis by Seurat 3, we identified 19 clusters identifiable as 8 cell types using known genetic markers: CNC-derived mesenchymal cells (*Meox1+*, *Dlx5+*), myogenic cells (*Myod1+*, *Myf5+*), neurons (*Tubb3+*, *Stmn2+*), endothelial cells (*Cdh5+*), erythroid cells (*Hba-x+*), glial cells (*Plp1+*, *Sox10+*), myeloid cells (*Lyz2+*), and epithelial cells (*Krt14+*) (*Figure 1A*; *Figure 1—figure supplement 1A*). Several lineages consisted of multiple clusters, such as CNC-derived mesenchymal, epithelial, neuronal, and myogenic cells, highlighting the heterogeneity within those populations (*Figure 1A*). Interestingly, in the CNC-derived mesenchymal cell population, we observed eight different clusters (Clusters 0–4, 7, 8, 10) (*Figure 1A–B*). Besides Clusters 2 and 10, which were identified as terminally differentiated osteogenic and chondrogenic cells, respectively, the cell types and functions of the other clusters in the CNC-derived mesenchymal population have not yet been well characterized.

To characterize the roles of these less known subpopulations, we analyzed the top 10 differentially expressed genes in each cluster and performed functional annotation for those highly specific markers using Ingenuity Pathway Analysis. We thus identified four major types of CNC-derived cells in the soft palate besides osteogenic (Cluster 2) and chondrogenic cells (Cluster 10) (*Figure 1B*). Cluster 0 was enriched with early CNC marker genes such as *Tfap2b*, *Six2*, and *Prrx1* (*Simões-Costa and Bronner, 2015*; *Soldatov et al., 2019*), so we suspected that this population might be an undifferentiated early progenitor population associated with early CNC cells, and accordingly we hypothesized that they were CNC-derived progenitors (*Figure 1B*). Genes enriched in Cluster 1 (*Tbx22*, *Wnt16*, *Meis2*) were associated with the palatal shelf midline during development (*Louw et al., 2015*; *Pauws et al., 2013*; *Warner et al., 2009*; *Figure 1B*; *Figure 1—figure supplement 1B*); hence, we refer to this cluster as midline mesenchymal cells. Clusters 3, 4, and 7 expressed high levels of genes related to head and muscle morphogenesis (*Cxcl12, Igf1, Aldh1a2*); we refer to them as perimysial cells (*Matt et al., 2008*; *Schiaffino and Mammucari, 2011*; *Vasyutina et al., 2005*; *Figure 1B*; *Figure 1—figure supplement 1C*). Interestingly, Cluster 8 was strongly enriched in genes associated with mitosis (*Top2a, Ccnb1, Ube2c*) (*Nielsen et al., 2020*; *Pines, 2011*; *Strauss et al., 2018*) even after cell cycle regression was performed (*Figure 1B*). We therefore refer to this cluster as mitotic cells.

To investigate the *in vivo* identities of each cluster, we performed RNAscope *in situ* hybridization of the soft palate at E14.5. Different soft palate myogenic sites develop sequentially from anterior to posterior direction. Specifically, in coronal sections, the unfused palatal shelves in the LVP region (posterior) protrude toward the midline and the myogenic cells grow in a lateral to medial direction along the palatal shelves at E14.5 (*Figure 1C*), while the palatal shelves in the TVP region (anterior) are already fused and the myogenic cells wrap around the pterygoid plate (*Figure 1D*). As the TVP and LVP myogenic sites are more identifiable than those of the PLG and PLP at E14.5, we used the former as reference points for the anatomical locations of each cluster in our analysis. Using the top enriched genes of each cluster, we identified their distinct anatomical locations *in vivo*. The *Ibsp+* osteogenic (*Figure 1E,M*) and *Col2a1+* chondrogenic clusters (*Figure 1F,N*) were mostly associated with part of the styloid process of the temporal bone in the LVP region and the pterygoid plate of the sphenoid bone in the TVP region. In the LVP region, the *Tfap2b+* progenitor cluster was mainly located in the lateral portions of the palatal shelves (*Figure 1G*). The majority of the *Aldh1a2+* perimysial cluster was distributed in the lateral portion while only a small portion of this cluster appeared in the central myogenic sites (*Figure 1H*). In contrast, the two other perimysial clusters (*Hic1+* and *Tbx15+*) were most abundantly located in the central myogenic sites of the LVP (*Figure 1I–J*). Midline mesenchymal cells (*Tbx22+*) were mainly located in the medial portions of the palatal shelves (*Figure 1K*). The *Top2a+* mitotic cells were distributed throughout the palatal shelves and adjacent to both early progenitors and committed CNC-derived cells (*Figure 1L*). A similar distribution of

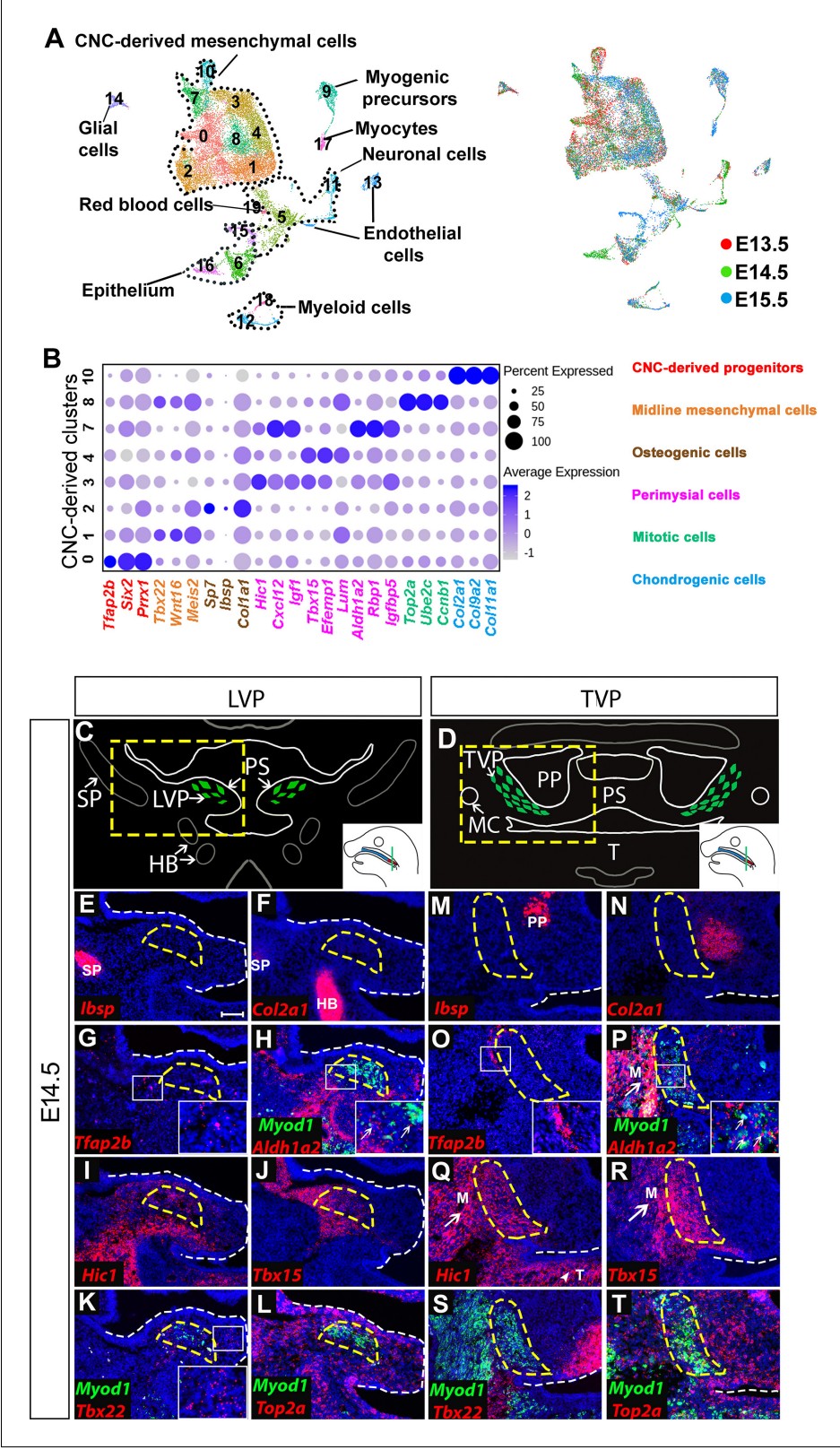

**Figure 1.** Analysis of distinct populations of cranial neural crest (CNC)-derived mesenchymal cells in the soft palate during development. (**A**) UMAP plot integration analysis of mouse soft palate cells from E13.5, E14.5, and E15.5 based on clusters (left) and different developmental stages (right). (**B**) DotPlot of signature genes in CNC-derived clusters. The color code of signature genes corresponds to the colors of the names of distinct cell populations in the right panel. (**C**) Schematic drawings of *Myod1* (green), styloid process of temporal bone (SP) and hyoid bone (HB) on coronal

*Figure 1 continued on next page*

*Figure 1 continued*

sections of the levator veli palatini (LVP) region and (D) *Myod1* (green), tongue (T), pterygoid plate (PP), and Meckel's cartilage (MC) on coronal sections of the tensor veli palatini (TVP) region of E14.5 control mice. PS, Palatal shelves. Yellow dashed boxes in (C) and (D) are enlarged and analyzed for expression patterns of cluster-specific markers in (E–L) and (M–T), respectively. (E–L) RNAscope *in situ* hybridization for *Myod1* and selected marker genes from each cluster of CNC-derived cells on coronal sections of the LVP region. (M–T) RNAscope *in situ* hybridization for *Myod1* and selected marker genes from each cluster of CNC-derived cells in the TVP region. White arrows point to masseter muscles (M) in (P, Q, R). White arrowhead points to tongue (T) in (Q). Yellow dashed lines outline the myogenic sites (LVP in E-L; TVP in M-T). White dashed lines outline the palatal shelf. Boxed areas are enlarged in the insets. Scale bar in E indicates 100 μm for E-T.

The online version of this article includes the following figure supplement(s) for figure 1:

**Figure supplement 1.** Identification and characterization of different cellular populations in the soft palate from E13.5 to E15.5.
**Figure supplement 2.** *Hic1* and *Tbx15* are expressed in cells surrounding other craniofacial muscles adjacent to the levator veli palatini (LVP) region.
**Figure supplement 3.** Expression of perimysial markers in the hard palate region.

different cluster markers was observed in the TVP region (*Figure 1O–T*). Outside of the soft palate, the perimysial markers (*Aldh1a2, Hic1* and *Tbx15*) were expressed in the mesenchyme surrounding the tongue and masseter muscles in addition to the palatal myogenic sites of the TVP (*Figure 1P–R*), while near the LVP region, *Hic1* and *Tbx15* were also expressed in the mesenchyme surrounding the migratory path of myogenic progenitors of the LVP and the myogenic sites of the middle pharyngeal constrictor muscle and tensor tympani muscle (*Figure 1I–J* and *Figure 1—figure supplement 2A–E*). These observations suggest the perimysial lineage might be a common CNC-derived sub-population involved in the development of multiple craniofacial muscles.

Because the oropharyngeal muscles are only present in the soft palate, not the hard palate, we investigated whether the perimysial markers are also specific to the soft palate. Interestingly, *Tbx15* expression was absent from the hard palate, but *Aldh1a2* and *Hic1* were expressed in the hard palate mesenchyme specifically surrounding the tooth germ (*Figure 1—figure supplement 3D–F*). This suggests that *Aldh1a2* and *Hic1* might have different functions in the hard and soft palate. Interestingly, we also observed that *Aldh1a2* was expressed in the medial mesenchyme of the tongue, while *Hic1* and *Tbx15* were expressed broadly in the mesenchyme of the tongue at E14.5 (*Figure 1—figure supplement 3C,E–F*). Moreover, the expression of *Aldh1a2* in the tongue gradually decreased from E12.5 to E14.5 (*Figure 1—figure supplement 3A–C*). As myogenic precursors started to appear in the center of tongue primordium (*Han et al., 2012*), the *Aldh1a2+* population might be specifically associated with early myogenic populations, while *Hic1+* and *Tbx15+* populations may be associated with more general myogenic populations.

## Runx2 is expressed in the perimysial populations and CNC-derived progenitor cells during soft palate development

To elucidate the dynamic process by which CNC-derived cells differentiate during soft palate development, we performed individual single-cell transcriptome analyses for E13.5, E14.5, and E15.5, then compared them. The pterygoid plate of the sphenoid bone and part of the styloid process of the temporal bone are not considered to be part of the palate, so we excluded the osteochondrogenic clusters belonging to these structures from further analysis. Interestingly, we observed decreased cell heterogeneity in CNC-derived soft palate mesenchymal populations during development. The number of CNC-derived clusters declined from seven at E13.5 to six at E14.5 and eventually five at E15.5 using the same unsupervised clustering settings (*Figure 2—figure supplement 1A*). In contrast, myogenic cells formed a single cluster at E13.5 but expanded to two clusters at E15.5 (*Figure 2—figure supplement 1A*).

To further investigate how each cluster changed over time, we extracted and compared the CNC-derived and myogenic cells from E13.5 to E15.5 based on the earlier integration analysis (*Figure 2A*). Consistent with previous observations, in the myogenic clusters we observed an increased number of both early myogenic precursors (Cluster 9, *Msc+, Myf5+*) and differentiated myocytes (Cluster 17, *Myl4+*) as development progressed (*Figure 2A*). The number of cells in Clusters 1 (*Tbx22+*), 3 (*Hic1+*) and 4 (*Tbx15+*) also increased from E13.5 to E15.5, but the number of cells in CNC-derived Clusters 0 (*Tfap2b+*), 7 (*Aldh1a2+*), and 8 (*Top2a+*) gradually decreased. This suggests that Clusters 0, 7, and 8 may be progenitors that are transiently present at early stages of soft palate development and give rise to Clusters 1, 3, and 4 (*Figure 2A–B*). To test this, we also

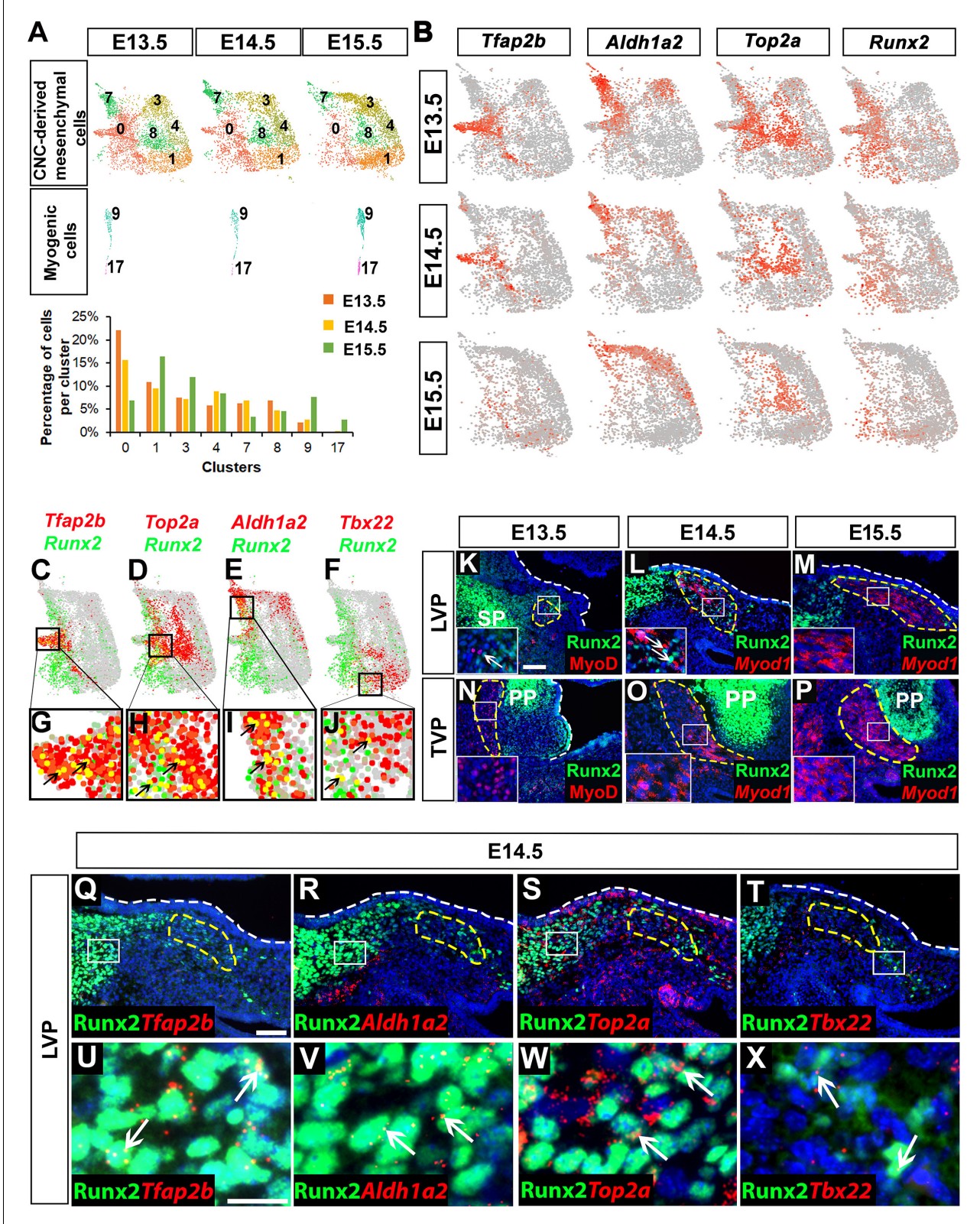

**Figure 2.** Runx2 is expressed in cranial neural crest (CNC)-derived progenitors and perimysial cells during soft palate development. (**A**) Individual UMAP clustering of CNC-derived mesenchymal and myogenic cells at three different embryonic stages (E13.5, E14.5, and E15.5) from integration analysis (top and middle panel). Percentages of cells in different CNC-derived and myogenic clusters in control soft palates at E13.5, E14.5, and E15.5 based on the integrated analysis (bottom panel). (**B**) Expression patterns of marker genes that are expressed transiently during early stages of soft

*Figure 2 continued on next page*

*Figure 2 continued*

palate development. (C–J) Co-expression of *Runx2* with cluster-specific markers *Tfap2b*, *Top2a*, *Aldh1a2*, *Tbx22* in E13.5-E15.5 soft palate integration analysis. Boxed areas in (C–F) are enlarged in (G–J). Black arrows point to cells co-expressing *Runx2* with individual cluster-specific markers. (K–P) Runx2 with myogenic markers MyoD or *Myod1* on coronal sections of the tensor veli palatini (TVP) and levator veli palatini (LVP) regions of control mice at E13.5, E14.5, and E15.5. Boxes indicate regions shown at higher magnification in the insets. (Q–X) Co-localization of Runx2 with cluster-specific marker genes *Tfap2b*, *Aldh1a2*, *Top2a*, and *Tbx22* on coronal sections of the LVP region of E14.5 control mice. Boxed areas in Q-T are enlarged in U-X. Yellow dashed lines in (K–T) outline the myogenic cells. White dashed lines outline the palatal shelf. Scale bars in K and Q indicate 100 µm for K-P and Q-T. Scale bar in U indicates 30 µm for U-X.

The online version of this article includes the following figure supplement(s) for figure 2:

**Figure supplement 1.** Cranial neural crest (CNC)-derived cells are heterogeneous and dynamic during soft palate development from E13.5 to E15.5.

computationally predicted the differentiation trajectory of CNC-derived cells using pseudotime analysis. Our results predicted Cluster 0 to be common CNC-derived progenitors that bifurcate into two more committed groups: perimysial progenitors (Cluster 7) for the later perimysial population (perimysial fibroblasts) (Clusters 3 and 4), and another group of progenitors (a subset of Cluster 0 and Cluster 8) for midline mesenchymal cells (Cluster 1) (*Figure 2—figure supplement 1B–C*). The integration analysis suggested that the fate decision between perimysial and midline mesenchymal cells happens at E13.5-E14.5. Cluster 8, predicted to be a more committed group of progenitors, represents the Top2a+ mitotic population. Because mitosis establishes a time window during which transcription factors can easily access and activate genes important for cell lineage determination (*Gurdon, 2016*), cells with high mitotic activity are likely undergoing cell fate transition. Therefore, those Top2a+ mitotic cells might be transitioning from early progenitor status to becoming more committed to a particular fate. Interestingly, the number of *Aldh1a2+* cells in Cluster 7 gradually decreased, but the number of *Aldh1a2+* cells increased in Clusters 3 and 4 from E13.5 to E15.5 (*Figure 2B*), probably because *Aldh1a2* labels both the majority of the early perimysial population (Cluster 7) and also some of the late perimysial populations (Clusters 3 and 4).

Notably, we observed that expression of *Runx2* in the CNC-derived mesenchyme gradually decreased from E13.5 to E15.5, suggesting it might play a role in regulating CNC-derived cell differentiation during early soft palate development (*Figure 2B*). Furthermore, *Runx2* was expressed not only by the CNC-derived common progenitors Cluster 0 (*Tfap2b+*), but also by other more committed progenitor cells, Cluster 8 (*Top2a+*) and Cluster 7 (*Aldh1a2+*), which were mainly distributed around the bifurcation regions of different lineages; only a few midline mesenchymal cells in Cluster 1 (*Tbx22+*) expressed *Runx2* (*Figure 2C–J*).

To investigate the functional significance of Runx2 for soft palate development *in vivo*, we examined Runx2 expression in the TVP and LVP regions of control mice. Double staining of Runx2 and the myogenic marker MyoD/*Myod1* from E13.5-E15.5 revealed changes in Runx2 expression in the myogenic region as development progressed. Runx2 expression was gradually restricted from most of the palate primordium at E13.5 to only the mesenchymal cells in the putative progenitor, perimysial, and osteogenic sites in the LVP region at E14.5; eventually, it was found only in the osteogenic regions at E15.5 (*Figure 2K–M*). As there was no detectable Runx2 expression in the TVP perimysial site from E13.5 to E15.5 (*Figure 2N–P*), we focused on the LVP region as we investigated the colocalization of Runx2 with markers of early CNC-derived progenitors and different lineages *in vivo*. Consistent with our single-cell analysis, Runx2 was predominantly expressed in the putative progenitor population (*Tfap2b+*), actively amplifying population (*Top2a+*) and perimysial cells (*Aldh1a2+*), with only a few in the midline mesenchymal cells (*Tbx22+*) (*Figure 2Q–X*). Previous studies have shown that CNC-derived cells guide the migration and potentially regulate the maturation of mesoderm-derived myogenic precursors in the soft palate through tissue-tissue interactions (*Grimaldi et al., 2015*; *Li et al., 2019*; *Sugii et al., 2017*). We hypothesized that Runx2 may regulate differentiation of CNC-derived cells in a cell-autonomous manner at early stages, which may indirectly affect myogenesis in the soft palate mesenchyme.

## Loss of *Runx2* in CNC-derived cells results in soft palate development defects

To test the functional significance of Runx2 in regulating soft palate muscle development, we specifically targeted Runx2 in CNC-derived palate mesenchymal cells. We first tested whether *Osr2-Cre*,

which specifically labels the CNC-derived cell subset in the developing palatal mesenchyme from the beginning of palatal shelf outgrowth (*Lan et al., 2007*), could also label the CNC-derived population in the soft palate. We confirmed in *Osr2-Cre;tdTomato* mice that tdTomato+ cells indeed contribute to soft palate mesenchyme including the perimysial cells surrounding all soft palatal muscles as early as E14.5 (*Figure 3—figure supplement 1A–G*). Furthermore, co-expression of tdTomato and Runx2 in the soft palate suggested that we could use *Osr2-Cre* to specifically delete *Runx2* in a subset of CNC-derived cells in the soft palate region (*Figure 3—figure supplement 1A–G*).

To test whether Runx2 is a key regulator of soft palate development, we next generated *Osr2-Cre;Runx2^{fl/fl}* mice, which showed cleft soft palate (5/10), misoriented muscle fibers and reduced muscle size (10/10) along with defects in hard tissues including the palatine bone (3/6) and pterygoid process (6/6) (*Figure 3A–F*; *Figure 3—figure supplement 2A–J*). Intraoral imaging and CT scans showed soft palate cleft in *Osr2-Cre;Runx2^{fl/fl}* mice (*Figure 3A–F*; *Figure 3—figure supplement 2D*). Notably, in analyzing the CT scans, we found that three out of six *Runx2* mutant mice with missing palatine bones and more severe pterygoid plate defects also had soft palate clefts, while the other three *Runx2* mutants without clefts exhibited palatine bones that were smaller, though not statistically significantly so, and less severe pterygoid plate defects, particularly shorter pterygoid plate height (*Figure 3—figure supplement 2A–J*), suggesting the severity of skeletal defects is associated with the variability of soft palate clefts in *Osr2-Cre;Runx2^{fl/fl}* mice. Consistent with the CT scans, histological analysis showed that the height of pterygoid plate was reduced and muscle attachment was abnormal in the TVP region of *Osr2-Cre;Runx2^{fl/fl}* mice (*Figure 3G–H,L–M*). Because the aponeurosis serves the important function of attaching the hard tissue to the muscle, we also analyzed the fibrous tendon tissue marked by *Scx* by RNAscope *in situ* hybridization. The tendon tissue did not extend to the midline in the palate primordium in the TVP region in *Osr2-Cre;Runx2^{fl/fl}* mice as it did in the controls at E16.5 (*Figure 3—figure supplement 3I–L*). It could be seen more clearly at P0 that the aponeurosis in *Osr2-Cre;Runx2^{fl/fl}* mice was thinner and it did not stretch from the lateral-oral side to the medial-nasal side as it did in control mice (*Figure 3—figure supplement 4A–D*), suggesting its attachment to the posterior bone probably was likely abnormal. As Runx2 is not expressed in the perimysial site of the TVP region, Therefore, this muscle attachment defect of the TVP might be due to disruption of the hard tissue and aponeurosis. In the LVP region, the muscles were reduced in size in *Osr2-Cre;Runx2^{fl/fl}* mice compared to controls (*Figure 3I–K,N–P*). Interestingly, a significant number of LVP muscle fibers had anterior-posterior alignment in *Osr2-Cre;Runx2^{fl/fl}* mice (*Figure 3O–P*), compared to the uniform lateral-medial alignment of LVP muscle fibers in controls (*Figure 3J–K*). This suggests that the muscle fibers were mis-oriented, similar to the phenotype seen in patients with cleft soft palate. Additionally, we observed that the muscle fibers had centralized nuclei in the soft palate of *Osr2-Cre;Runx2^{fl/fl}* mice, which suggests that they had muscle differentiation defects (*Figure 3P*). Similar muscle defects were also observed in other palatal muscles such as the PLP (*Figure 3—figure supplement 3C–D,G–H*). We concluded that loss of *Runx2* leads directly to defects in CNC-derived cells and indirectly to muscle defects in the soft palate.

To examine soft palate muscle differentiation in *Osr2-Cre;Runx2^{fl/fl}* mice, we analyzed expression of myogenic markers at multiple developmental stages to identify the time point at which muscle defects began to appear using the LVP as an example. In the LVP, there was no apparent change of early myogenic marker MyoD expression between control and *Osr2-Cre;Runx2^{fl/fl}* mice at E13.5 (*Figure 4—figure supplement 1A–B*). MyoD staining revealed that defects started to appear at E14.5 (*Figure 4A–B*), when the palatal shelves began to grow and protrude towards the midline. Expression of the late myogenic marker MHC was decreased in the soft palate of *Osr2-Cre;Runx2^{fl/fl}* mice compared to controls at E15.5 (*Figure 4C–D*), suggesting delayed muscle differentiation. This reduced expression of MHC persisted in the soft palate of *Osr2-Cre;Runx2^{fl/fl}* mice at E16.5 (*Figure 4—figure supplement 1C–D*). MHC staining suggested that the myoblasts had fused to form myofibers, which were uniformly aligned in layers running in the lateral-to-medial direction in the LVP of control samples at E15.5 (*Figure 4C*). However, more immature myoblasts and fewer differentiated myofibers were present in *Osr2-Cre;Runx2^{fl/fl}* mice (*Figure 4D*) and the myofibers extended in different directions, potentially hindering further muscle development and compromising physiological function.

To investigate the cellular mechanism underlying soft palate defects in *Osr2-Cre;Runx2^{fl/fl}* mice, we analyzed cell proliferation, apoptosis, and differentiation. Consistent with the MyoD expression

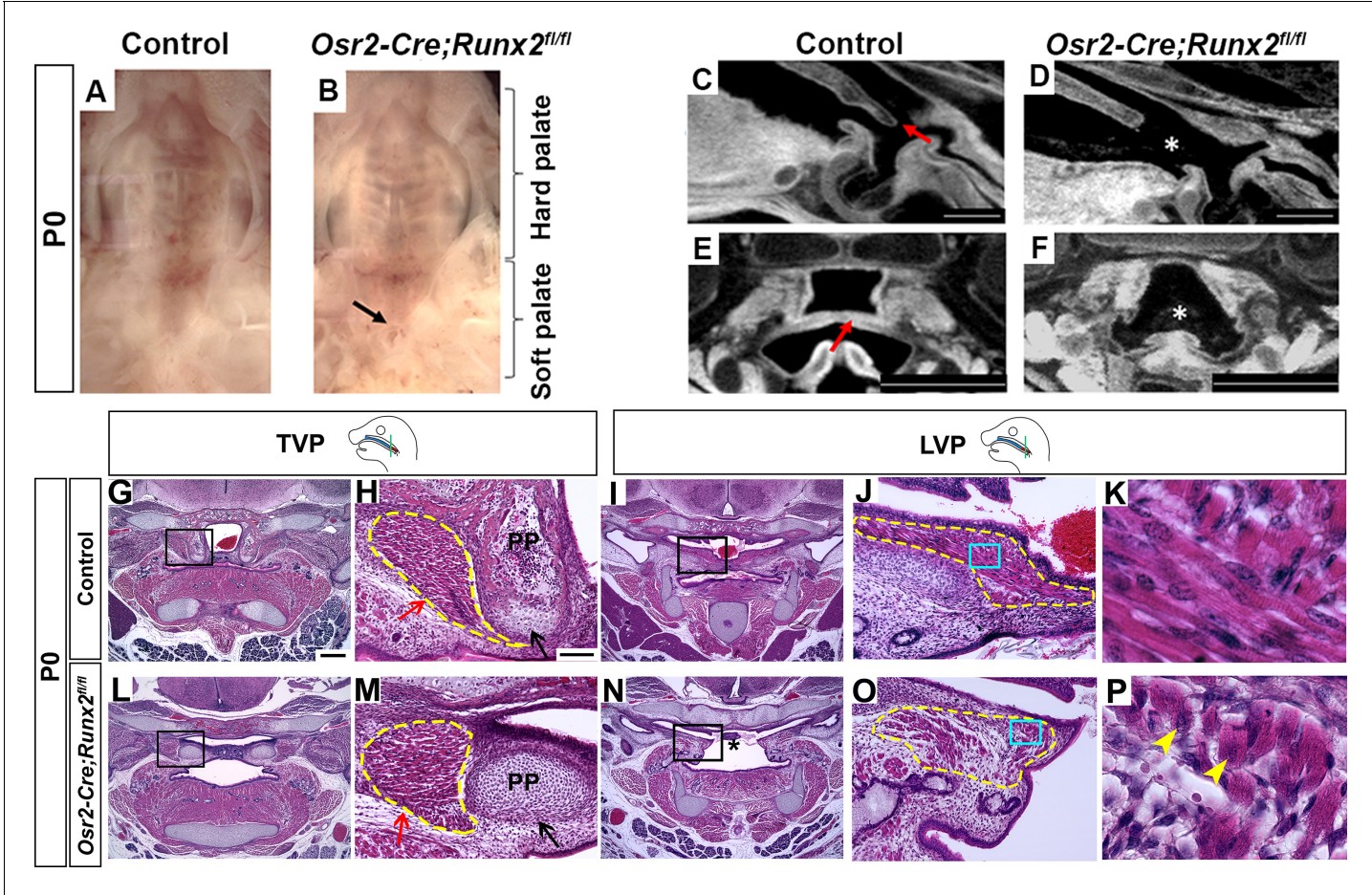

**Figure 3.** Deletion of *Runx2* in cranial neural crest (CNC)-derived cells leads to craniofacial defects in the soft palate. (**A, B**) Intraoral views of palates from control and *Osr2-Cre;Runx2^{fl/fl}* mice at newborn stage (P0). Arrow indicates the cleft in the posterior part of the soft palate. (**C–F**) Sagittal (**C–D**) and coronal (**E–F**) views of microCT scans of newborn control and *Osr2-Cre;Runx2^{fl/fl}* mice (N = 3). Red arrows indicate the normal soft palate, and asterisks indicate the cleft in the posterior part of soft palate. (**G–P**) H and E staining of soft palate coronal sections from control and *Osr2-Cre;Runx2^{fl/fl}* mice at P0 (N = 5). Yellow dashed lines outline the soft palate muscles. Black and red arrows in H and M show the pterygoid plate and tensor veli palatini (TVP) defects, respectively, of *Osr2-Cre;Runx2^{fl/fl}* mice. Asterisks in N indicate the cleft soft palate in the levator veli palatini (LVP) region of *Osr2-Cre;Runx2^{fl/fl}* mice. Boxed areas in G, I, L, and N are enlarged in H, J, M, and O, respectively. Boxed areas in J and O are enlarged in K and P, respectively. Scale bars in C-D and E-F indicate 0.5 mm and 0.9 mm, respectively. Scale bar in G indicates 400 μm for G, I, L, and N. Scale bar in H indicates 100 μm for H, J, M, and O. Yellow arrowheads in P indicate the centralized nuclei in mutant muscle cells.

The online version of this article includes the following source data and figure supplement(s) for figure 3:

**Figure supplement 1.** *Osr2-Cre* specifically deletes *Runx2* in the soft palate region.

**Figure supplement 2.** Deletion of *Runx2* in cranial neural crest (CNC)-derived cells gives rise to hard tissue defects in *Osr2-Cre;Runx2^{fl/fl}* mice.

**Figure supplement 2—source data 1.** Source data for *Figure 3—figure supplement 2I*.

**Figure supplement 2—source data 2.** Source data for *Figure 3—figure supplement 2J*.

**Figure supplement 3.** Deletion of *Runx2* in cranial neural crest (CNC)-derived cells gives rise to soft palate defects in *Osr2-Cre;Runx2^{fl/fl}* mice.

**Figure supplement 4.** Deletion of *Runx2* in cranial neural crest (CNC)-derived cells gives rise to aponeurosis defects in *Osr2-Cre;Runx2^{fl/fl}* mice.

pattern, we did not detect any change in the number of BrdU+ proliferating cells in the LVP region of *Osr2-Cre;Runx2^{fl/fl}* mice compared to controls at E13.5 (*Figure 4—figure supplement 1A–B*). In the LVP region at E14.5 and E15.5, the proliferation rate of MyoD- CNC-derived cells did not have significant difference in the perimysial sites of *Osr2-Cre;Runx2^{fl/fl}* mice compared to controls (*Figure 4E–H,I,K*). The proliferation rate of MyoD+ myogenic cells was not significantly different between controls and *Runx2* mutants at E14.5 (*Figure 4E–F,J*), but a significant reduction in the proliferation rate was observed in *Runx2* mutants at E15.5 (*Figure 4G–H,L*). We also performed caspase3 immunofluorescence staining to investigate cell apoptosis. The number of apoptotic cells was

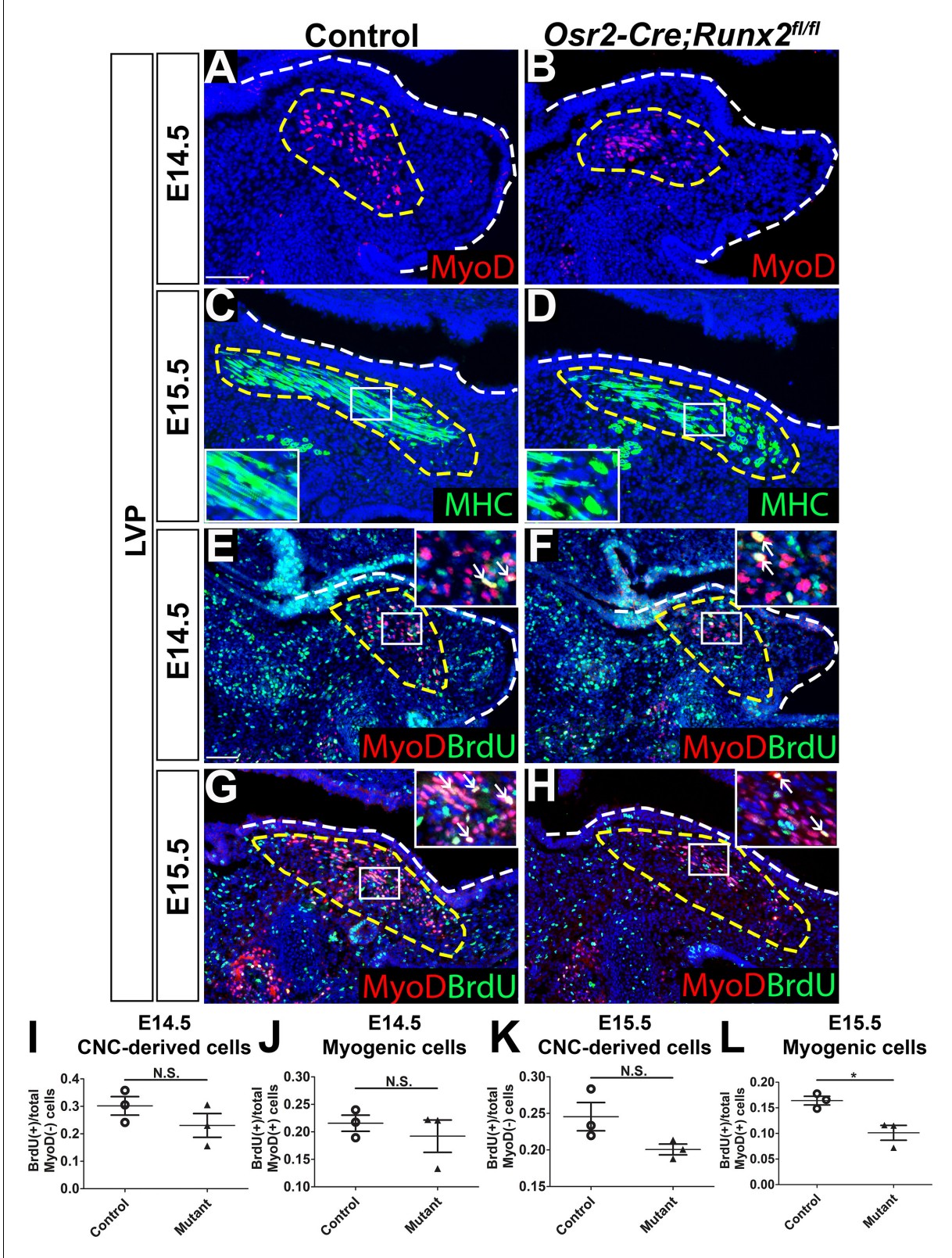

**Figure 4.** Loss of *Runx2* in cranial neural crest (CNC)-derived cells results in myogenic proliferation and differentiation defects of the levator veli palatini (LVP). (**A–B**) MyoD and (**C–D**) MHC immunostaining on coronal sections of the LVP regions of control and *Osr2-Cre;Runx2^{fl/fl}* mice at E14.5 and E15.5. Yellow dashed lines outline the myogenic cells. Boxed areas are enlarged as insets in the same image. (**E–H**) Immunostaining of BrdU and MyoD on coronal sections from the LVP regions of control and *Osr2-Cre;Runx2^{fl/fl}* mice at E14.5 and E15.5. Yellow dashed lines outline the location of

*Figure 4 continued on next page*

*Figure 4 continued*

myogenic cells in the LVP regions. Boxed areas are enlarged as insets in the same image. White arrows in the insets indicate BrdU+ myogenic cells. (I–L) Quantitation of proliferation rates of CNC-derived and myogenic cells in E14.5 (I–J) and E15.5 (K–L) coronal sections of the LVP regions of control (E, G) and *Osr2-Cre;Runx2fl/fl* (F, H) mice (N = 3 mice, four sections per region per mouse). White dashed lines outline the palatal shelf. * indicates p value = 0.02. Scale bars in A and E indicate 100 μm for A-D and E-H, respectively.

The online version of this article includes the following source data and figure supplement(s) for figure 4:

**Source data 1.** Source data for *Figure 4I–L*.

**Figure supplement 1.** Loss of *Runx2* in cranial neural crest (CNC)-derived cells results in myogenic differentiation but no apoptotic defects of the levator veli palatini (LVP).

---

indistinguishable between controls and mutants at E14.5 and E15.5 (*Figure 4—figure supplement 1E–H*). It is worth noting that although the proliferation rate of CNC-derived cells in the *Runx2* mutant mice were not significantly different from that of the controls, we observed that they had fewer and less proliferative MyoD+ cells than *Runx2fl/fl* control mice. These differences might be due to altered signaling in CNC-derived cells causing the reduction of MyoD expression as well as proliferation defects of myogenic cells in *Osr2-Cre;Runx2fl/fl* mice.

## Runx2 plays an important role in the lineage commitment of CNC-derived cells in the soft palate

To investigate whether Runx2 regulates CNC-derived cell fate determination during soft palate development, we compared cell composition and gene expression profiles of E14.5 *Osr2-Cre;Runx2fl/fl* and control soft palates using scRNA-seq, bulk RNA-seq, and *in vivo* expression analyses. Using integration analysis based on shared variance, we identified similar cell clusters in the soft palates of control and *Osr2-Cre;Runx2fl/fl* mice at this stage. However, the composition of the CNC-derived cells was altered in the *Runx2* mutants compared to controls (*Figure 5A*). Using markers of different subtypes of CNC-derived cells, we observed that the percentage of perimysial cells (Cluster 4) in the population decreased in *Runx2* mutants, while the percentage of midline mesenchymal cells (Clusters 0 and 3) increased (*Figure 5B*). Moreover, *in situ* RNAscope staining revealed decreased expression of perimysial markers in the soft palates of *Runx2* mutants compared to controls at E14.5, suggesting the CNC-derived perimysial populations were affected, potentially leading to further myogenic defects (*Figure 5—figure supplement 1A–H*). Consistent with the scRNA-seq results, bulk RNA-seq also identified that certain genes associated with specific types of CNC-derived cells were differentially expressed in the *Osr2-Cre;Runx2fl/fl* mice (*Figure 5—figure supplement 1I*). A number of genes not exclusively associated with specific types of CNC-derived cells, including *Twist1 and Meox2*, were also identified as being differentially expressed in the bulk RNA-seq analysis (*Figure 5—figure supplement 1I*).

We focused our attention on Twist1, which inhibits binding of Runx2 to its downstream targets and antagonizes Runx2's function in osteoblasts (*Bialek et al., 2004*). We began by analyzing the expression pattern of *Twist1* during soft palate development. Based on the integration analysis of E13.5-E15.5 single-cell transcriptomes from controls, we observed *Twist1* was primarily expressed in midline mesenchymal cells (*Tbx22+*), while its expression in CNC-derived common progenitors and perimysial cells (*Tfap2b+*; *Aldh1a2+*) was relatively low (*Figure 5—figure supplement 2A*). In addition, expression of *Twist1* in the CNC-derived cells changed over time. In the LVP region, at E13.5 *Twist1* was expressed at a low level in the palate primordium and perimysial sites (*Figure 5—figure supplement 2B–C*). At E14.5, expression of *Twist1* in the palate primordium had increased, whereas its expression was maintained at a low level in the perimysial site (*Figure 5—figure supplement 2D–E*). At E15.5, *Twist1* showed a similar expression pattern to that of E14.5 (*Figure 5—figure supplement 2F–G*). This spatiotemporally specific *Twist1* expression in the palate primordium and myogenic regions of the soft palate was accompanied by an opposite trend in Runx2 expression in the same regions at the same stages. This is perhaps shown most clearly by the colocalization of Runx2 and *Twist1* in the LVP region at E14.5 (*Figure 5D–G*), which suggested that expression levels of Runx2 and *Twist1* are tightly coordinated during soft palate development. Interestingly, we discovered that expression of *Twist1* was upregulated in most of the palatal shelf region including the

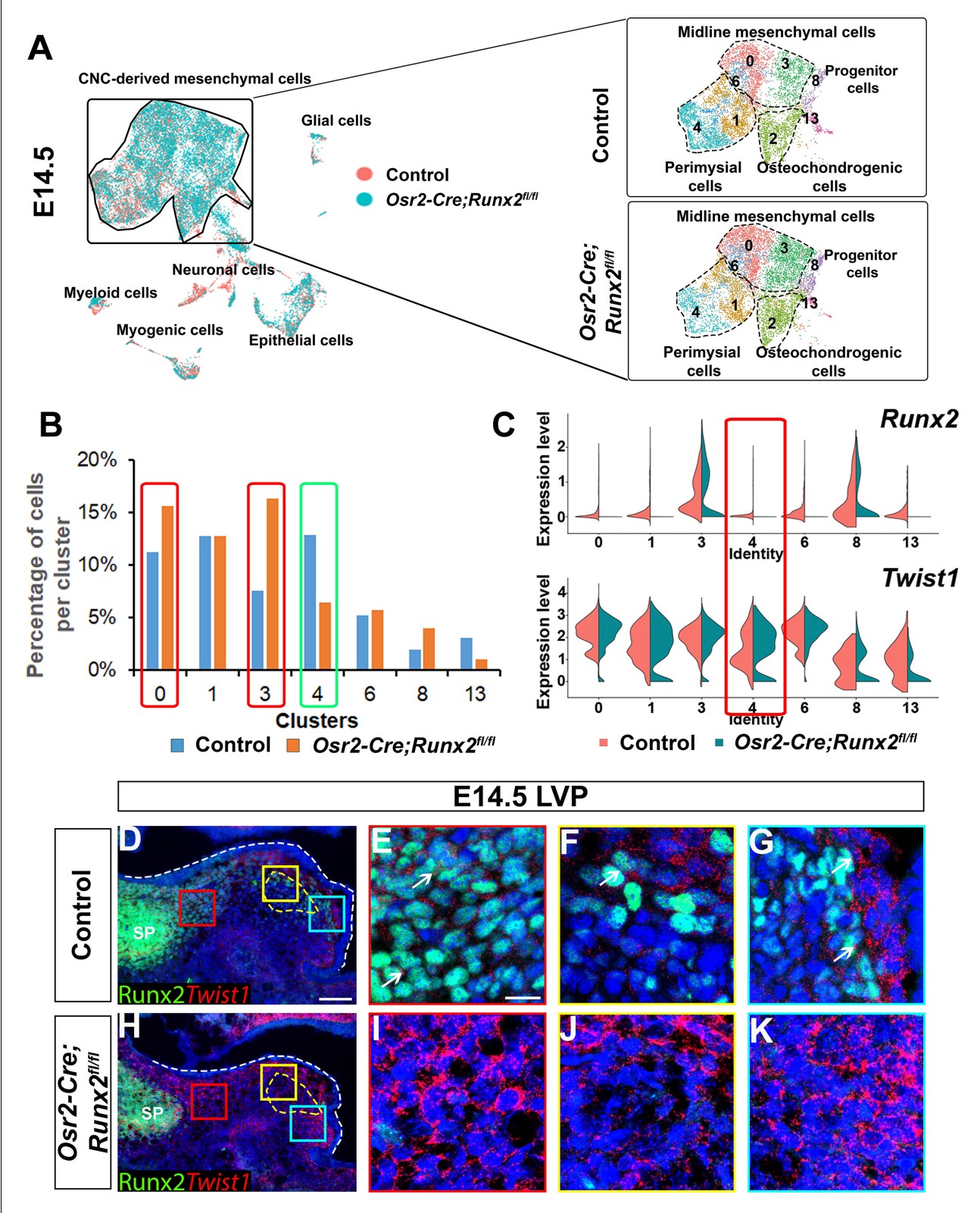

**Figure 5.** Loss of *Runx2* results in altered cell fate of cranial neural crest (CNC)-derived mesenchymal cells in the soft palate. (**A**) Integration analysis of the soft palate regions of control and *Osr2-Cre;Runx2^{fl/fl}* mice at E14.5. Left panel shows the integration analysis of control and *Osr2-Cre;Runx2^{fl/fl}* at E14.5. Right panels show the split UMAP clustering views of CNC-derived mesenchymal cells from control and *Osr2-Cre;Runx2^{fl/fl}* mice based on the integration analysis. Black dotted lines outline the different subtypes of CNC-derived mesenchymal cells in the soft palate. (**B**) Percentages of cells in

*Figure 5 continued on next page*

Figure 5 continued

different CNC-derived non-osteochondrogenic clusters in control and *Osr2-Cre;Runx2^{fl/fl}* soft palates based on the integration analysis in (A). Red boxes and green boxes indicate the clusters with an increased and decreased percentages of cells, respectively, in *Osr2-Cre;Runx2^{fl/fl}* mice compared to controls. (C) Violin plots show the comparative expression levels of *Runx2* and *Twist1* in different CNC-derived non-osteochondrogenic clusters. Red box highlights the differences in *Runx2* and *Twist1* expression in perimysial cell clusters. (D–K) Co-expression of Runx2 and *Twist1* on coronal sections of the levator veli palatini (LVP)regions of control and *Osr2-Cre;Runx2^{fl/fl}* mice at E14.5. Yellow dashed lines outline the myogenic sites. Red, yellow and blue boxes in D and H are enlarged in E, F, G and I, J, K, respectively. White dashed lines outline the palatal shelf. Scale bar in D indicates 100 μm for D and H. Scale bar in E indicates 20 μm for E-G and I-K.

The online version of this article includes the following figure supplement(s) for figure 5:

**Figure supplement 1.** Expression of lineage-specific and general markers is altered in *Osr2-Cre;Runx2^{fl/fl}* mice.

**Figure supplement 2.** *Twist1* is expressed in the soft palate in a spatiotemporally specific manner.

perimysial cells in *Osr2-Cre;Runx2^{fl/fl}* mice (*Figure 5H–K*), which suggests that upregulation of *Twist1* in the CNC-derived cells may interrupt their fate determination.

## Haploinsufficiency of *Twist1* rescues soft palate defects in *Osr2-Cre; Runx2^{fl/fl}* mice

Based on the complimentary expression patterns of Runx2 and *Twist1* in the soft palate, we hypothesized that they may oppose each other in regulating their common downstream targets which are important for fate determination of CNC-derived cells. Therefore, we performed ATAC-seq and found that both Runx2 and Twist1 binding sites are present in the regulatory region located around 15–40 kb downstream of the genetic locus of perimysial marker *Aldh1a2* (*Figure 6A*), suggesting that both Runx2 and Twist1 might directly regulate the expression of *Aldh1a2*. However, as the binding sites of those two transcription factors are more than 20 kb apart, they are likely to regulate *Aldh1a2* independently. Based on this finding, we sought to investigate whether haploinsufficiency of *Twist1* may rescue the soft palate defects in *Osr2-Cre;Runx2^{fl/fl}* mice by generating *Osr2-Cre; Runx2^{fl/fl};Twist1^{fl/+}* mice. Histological analysis confirmed that the palatal stromal mesenchyme, pterygoid plate, and muscle defects were all indeed rescued in these mice. None of the five newborn *Osr2-Cre;Runx2^{fl/fl};Twist1^{fl/+}* pups we collected had palatal clefts, compared to the 50% penetrance of cleft soft palate in *Osr2-Cre;Runx2^{fl/fl}* mice (*Figure 6B–G*). Pterygoid plate height was restored in *Osr2-Cre;Runx2^{fl/fl};Twist1^{fl/+}* mice (*Figure 6B–D,H*), and muscle fiber orientation and muscle size were also recovered (*Figure 6E–G*). To confirm whether the rescue of these muscle defects was due to restoration of perimysial genes, we performed *in situ* hybridization expression analysis of the soft palate of *Osr2-Cre;Runx2^{fl/fl};Twist1^{fl/+}* mice at E14.5 (*Figure 6J–R*). *Aldh1a2* was downregulated in the region (*Figure 6K*) where *Runx2* was deleted (*Figure 6N*) and *Twist1* expression was expanded in the soft palate of *Osr2-Cre;Runx2^{fl/fl}* mice (*Figure 6Q*). Compared to *Osr2-Cre;Runx2^{fl/fl}* mice, the expression of *Aldh1a2* was restored in the *Osr2-Cre;Runx2^{fl/fl};Twist1^{fl/+}*mice at E14.5 (*Figure 6L*). This suggests that Runx2 and Twist1 exhibit opposite regulatory effects on the expression of *Aldh1a2* in a subset of CNC-derived cells, which may be important for regulating muscle differentiation in the soft palate. Since haploinsufficiency of *Twist1* in *Osr2-Cre;Runx2^{fl/fl};Twist1^{fl/+}* mice rescues the expression of *Aldh1a,* it is most likely that Runx2 activates the expression of *Aldh1a2* through repressing *Twist1* instead of directly activating *Aldh1a2* during normal soft palate muscle development (*Figure 6—figure supplement 1*).

## Discussion

CNC-derived cells are essential for craniofacial musculoskeletal development, as they give rise to multiple hard and soft tissues in the system, and guide muscle development (*Heude et al., 2011*; *Sugii et al., 2017*; *Tzahor, 2015*). These multiple roles are likely achieved by different subtypes of CNC-derived cells. The heterogeneity of CNC-derived cells has long been studied in the palate based on its anatomical structures along the anterior-posterior, mediolateral, and oral-nasal axes (*Bush and Jiang, 2012*; *Han et al., 2009*; *Li et al., 2017*; *Potter and Potter, 2015*). To date, understanding of the molecular heterogeneity in different regions of the soft palate mesenchyme has mainly been based on location-specific genes that control local development and signal induction during palate outgrowth (*Han et al., 2009*; *Potter and Potter, 2015*). However, this does not

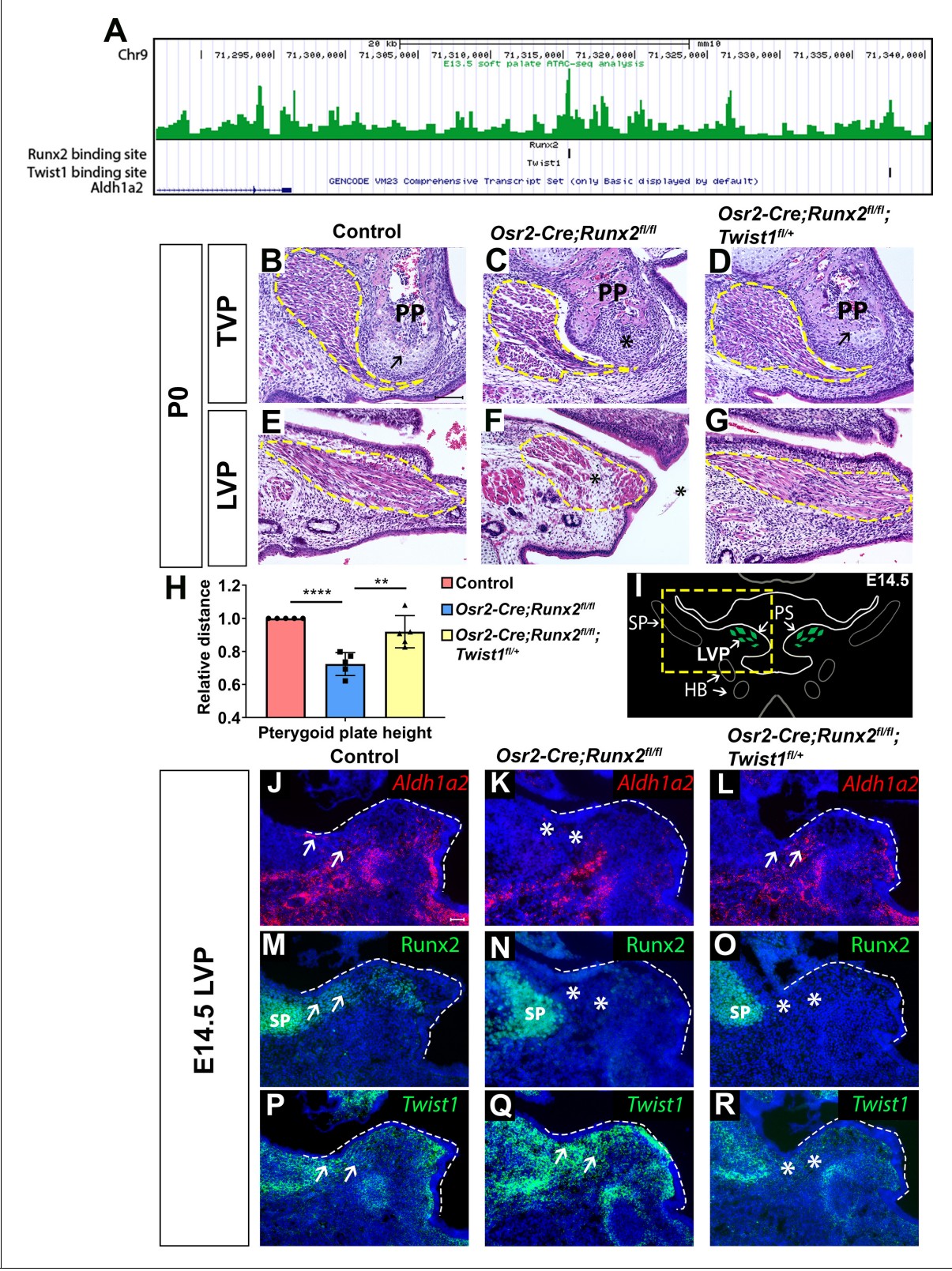

**Figure 6.** Twist1 antagonizes Runx2 to regulate soft palate development. (**A**) ATAC-seq peaks showing Twist1- and Runx2-binding sites are present in the opened regulatory regions near the *Aldh1a2* locus in the soft palate tissue. (**B–G**) H and E staining of tensor veli palatini (TVP) and levator veli palatini (LVP) coronal sections in P0 control, *Osr2-Cre;Runx2^{fl/fl}* and *Osr2-Cre;Runx2^{fl/fl};Twist1^{fl/+}* mice (N = 5). Yellow dashed lines outline the location of myogenic cells. Arrows indicate comparable structures in the pterygoid plates (PP) of control and *Osr2-Cre;Runx2^{fl/fl};Twist1^{fl/+}* mice. Asterisk indicates

*Figure 6 continued on next page*

*Figure 6 continued*

defective pterygoid plate, palate and LVP muscles in *Osr2-Cre;Runx2$^{fl/fl}$*. (H) Quantification of the height of the palatine bone from control (red bars), *Osr2-Cre;Runx2$^{fl/fl}$* (blue bars) mice and *Osr2-Cre;Runx2$^{fl/fl}$;Twist1$^{fl/+}$* (yellow bars) mice (N = 5). (I) Schematic drawings of *Myod1* (green), styloid process of temporal bone (SP) and hyoid bone (HB) on coronal sections in the LVP region of E14.5 control mice. (J–R) *Aldh1a2* RNAscope *in situ* hybridization (J–L), Runx2 immunostaining (M–O) and *Twist1* RNAscope *in situ* hybridization (P–R) in E14.5 LVP coronal sections of control, *Osr2-Cre;Runx2$^{fl/fl}$* and *Osr2-Cre;Runx2$^{fl/fl}$;Twist1$^{fl/+}$* mice. White dashed lines outline the palatal shelf. Scale bars in B and J indicate 100 µm for B-G and J-R, respectively. The online version of this article includes the following source data and figure supplement(s) for figure 6:

**Source data 1.** Source data for *Figure 6H*.

**Figure supplement 1.** Schematic depicts our model of the mechanism of soft palate development in control (left panel) and *Osr2-Cre;Runx2$^{fl/fl}$* mice (right panel).

completely explain how molecular heterogeneity contributes to the multiple roles played by CNC-derived cells during palate formation, or specifically how they guide soft palate muscle development.

In this study, we have revealed the cellular-level heterogeneity in the soft palate and established that different subtypes of CNC-derived cells are associated with distinct differentiation potentials and functions. Functional analysis of each CNC-derived cluster shows previously unknown subtypes of CNC-derived cells, and computational analysis suggests previously unknown lineage differentiation trajectories. *Tfap2b+* cells are the least differentiated subtype and give rise to *Aldh1a2+* perimysial progenitor cells and *Top2a+* transitioning cells, which further differentiate into perimysial fibroblasts and midline mesenchymal fibroblasts. *Tfap2b+* common progenitors, transitioning cells and some *Aldh1a2+* perimysial progenitor cells are transiently present only at early stages of soft palate development, consistent with their roles as progenitors. This transient presence of CNC-derived progenitor cells is similar to that of neural crest cells, which are known to be pluripotent during embryonic development and disappear at later stages (*Bronner and LeDouarin, 2012*). In addition, we show that markers labeling soft palate perimysial populations, such as *Hic1*, *Aldh1a2*, and *Tbx15*, are also expressed by connective tissues in other craniofacial muscles, including the tongue and masseter muscles. Consistent with our findings, *Aldh1a2* is known as an important enzyme for retinoic acid signaling, which is crucial for neural crest cells as they guide the positioning of extraocular muscles (*Matt et al., 2008*). The potential myogenic-supportive function of *Hic1* has been confirmed by the recent finding that the Hic1+ population represents a source of quiescent mesenchymal progenitors that play important roles during the regeneration of skeletal muscles in the limbs (*Scott et al., 2019*). Hence, *Aldh1a2* and *Hic1* might be novel markers for CNC-derived perimysial tissues, which may perform important pro-myogenic functions during muscle development.

During embryonic development, the palatal shelves grow in a lateral-to-medial direction both before and after their elevation (*Bush and Jiang, 2012*). Consistent with this, our *in vivo* analysis shows that the least differentiated *Tfap2b+* subpopulation is located in the lateral region of the soft palate during the early stages of its development, and the perimysial and midline mesenchymal populations reside in central myogenic sites and the medial region of the soft palate. Our results have revealed complex cellular heterogeneity and a differentiation hierarchy of cell populations that contribute to the craniofacial musculoskeletal system, which will require further analysis.

Recently, studies have found that several transcription factors regulate the development of different components of a musculoskeletal complex in a coordinated fashion to form a functional unit (*Colasanto et al., 2016*; *Hasson et al., 2010*; *Mathew et al., 2011*; *Vickerman et al., 2011*). Our study shows that Runx2 is expressed in CNC-derived cells involved in early cell fate determination and in perimysial cells in the soft palate mesenchyme. Loss of *Runx2* in CNC-derived cells of the soft palate mesenchyme leads to multiple tissue defects in the soft palate, including fibrous tendon tissue, soft palate cleft and muscle defects. There is a fate change of CNC-derived cells from perimysial cells to midline mesenchymal cells in the soft palate of *Runx2* mutant mice. As perimysial cells are closely associated with muscle development, loss of *Runx2* affecting their differentiation may in turn affect their secretion of signaling cues that promote muscle proliferation and differentiation. Indeed, we show that multiple genes associated with pro-myogenic secreted factors specifically expressed by perimysial populations, such as *Aldh1a2*, *Igf1*, *Cxcl12*, and *Cthrc1*, are downregulated in *Runx2* mutant mice (*Matt et al., 2008*; *Schiaffino and Mammucari, 2011*; *Spector et al., 2013*;

*Vasyutina et al., 2005*). Our study provides clues as to how those transcription factors might play different roles in regulating multiple musculoskeletal system components, but how the development of multiple components is integrated still needs further investigation.

Transcription factors often regulate different downstream targets in distinct tissues. Previous studies have shown that Runx2 regulates the differentiation of CNC-derived cells during early tooth and intramembranous bone formation through distinct sets of downstream targets including *Gli1*, *Lef1*, *Tcf1*, *Wnt10a*, *Wnt10b*, and *Tgfb1* in osteogenic cells and *Dusp6*, *Enpp1*, *Igfbp3*, and *Fgf3* in dental mesenchyme (*James et al., 2006*). In this study, we have shown that downstream targets of Runx2 have differing responses to the loss of *Runx2* in the soft tissue. Perimysial markers were specifically downregulated upon loss of Runx2, while genes expressed specifically in the midline mesenchymal cells and a set of more broadly expressed genes are upregulated. Although Runx2 has both transcriptional activation and repression domains, its different regulatory effects on distinct downstream targets in the soft palate mesenchyme could be direct or indirect. Our results thus reveal previously unknown roles of Runx2 in muscle development and help to elucidate the tissue-specific regulatory mechanisms by which Runx2 guides development.

Twist1 suppresses the function of its binding partner Runx2 through blocking the DNA binding domain of Runx2 to inhibit osteoblast differentiation and promote chondrocyte maturation (*Bialek et al., 2004*; *Hinoi et al., 2006*). In this study, we reveal complimentary expression patterns of Runx2 and *Twist1* in the perimysial and midline mesenchymal populations during soft palate development, which seems to confirm their antagonistic interaction (*Bialek et al., 2004*). However, in contrast to the previously reported model of Twist1 and Runx2 antagonistic interaction, we show that loss of *Runx2* in CNC-derived cells leads to abnormal upregulation of *Twist1* in the perimysial population. Further analysis has shown that suppression of *Twist1* in the perimysial population by Runx2 is necessary to maintain the expression level of perimysial marker gene *Aldh1a2*, which may be important for regulating muscle development. Our findings thus reveal a novel mechanism of Runx2-Twist1 genetic interaction that integrates the development of different types of CNC-derived cells with muscles to guide them to form a functional unit in the soft palate.

In summary, our study reveals a complex cellular heterogeneity within the developing soft palate and demonstrates that distinct subpopulations of CNC-derived cells are associated with distinct functions, which coordinate to form intricately connected components of the oropharyngeal complex. Moreover, the regulation of myogenesis by perimysial CNC-derived cells through Runx2-Twist1 interaction in the soft palate might also be shared by other craniofacial musculoskeletal structures. Our study highlights the complex regulatory roles of CNC-derived cells in the development of craniofacial musculoskeletal systems and provides knowledge that may lead to new strategies for craniofacial muscle regeneration.

# Materials and methods

## Key resources table

| Reagent type (species) or resource | Designation | Source or reference | Identifiers | Additional information |
|---|---|---|---|---|
| Strain, strain background (*M. musculus*) | *Runx2flox/flox* | *Takarada et al., 2013* | | |
| Strain, strain background (*M. musculus*) | *Twist1flox/flox* | *Bildsoe et al., 2009* | RRID:MMRRC_016842-UNC | |
| Strain, strain background (*M. musculus*) | *ROSA26loxp-STOP -loxp-tdTomato* | Jackson Laboratory | Stock No. 007905; RRID:IMSR_JAX:007905 | |
| Strain, strain background (*M. musculus*) | *Osr2-Cre* | Rulang Jiang, Cincinnati Children's Hospital | | |
| Sequence-based reagent | Mm-Myod1 probe | Advanced Cell Diagnostics | Cat# 316081 | |

*Continued on next page*

*Continued*

| Reagent type (species) or resource | Designation | Source or reference | Identifiers | Additional information |
|---|---|---|---|---|
| Sequence-based reagent | Mm-Scx probe | Advanced Cell Diagnostics | Cat# 439981 | |
| Sequence-based reagent | Mm-Twist1 probe | Advanced Cell Diagnostics | Cat# 414701 | |
| Sequence-based reagent | Mm-Aldh1a2 probe | Advanced Cell Diagnostics | Cat# 447391 | |
| Sequence-based reagent | Mm-Hic1 probe | Advanced Cell Diagnostics | Cat# 464131 | |
| Sequence-based reagent | Mm-Tfap2b probe | Advanced Cell Diagnostics | Cat# 536371 | |
| Sequence-based reagent | Mm-Tbx22 probe | Advanced Cell Diagnostics | Cat# 426511 | |
| Sequence-based reagent | Mm-Ibsp probe | Advanced Cell Diagnostics | Cat# 415501 | |
| Sequence-based reagent | Mm-Col2a1 probe | Advanced Cell Diagnostics | Cat# 407221 | |
| Sequence-based reagent | Mm-Tbx15 probe | Advanced Cell Diagnostics | Cat# 558761 | |
| Sequence-based reagent | Mm-Top2a probe | Advanced Cell Diagnostics | Cat# 491221 | |
| Sequence-based reagent | Mm-tdTomato probe | Advanced Cell Diagnostics | Cat# 317041 | |
| Antibody | Rabbit monoclonal anti-Runx2 | Cell Signaling Technology | RRID:AB_2732805 Cat# 12556S | (1:100) |
| Antibody | Rabbit monoclonal anti-active Caspase 3 | Cell Signaling Technology | RRID:AB_2341188 Cat# 9661S | (1:100) |
| Antibody | Rat monoclonal anti-BrdU | Abcam | RRID:AB_305426 Cat# ab6326 | (1:100) |
| Antibody | Mouse monoclonal anti-MyoD | DAKO | RRID:AB_2148874 Cat# M3512 | (1:20) |
| Antibody | Mouse monoclonal anti-MHC | DSHB | Cat# P13538 | (1:10) |
| Antibody | Goat polyclonal anti-Mouse Alexa Fluor 488 | Life Technologies | RRID:AB_2534069 Cat# A11001 | (1:200) |
| Antibody | Goat polyclonal anti-Mouse Alexa Fluor 568 | Life Technologies | RRID:AB_2534072 Cat# A-11004 | (1:200) |
| Antibody | Goat polyclonal anti-Rat Alexa Fluor 488 | Life Technologies | RRID:AB_141373 Cat# A-11006 | (1:200) |
| Antibody | Goat polyclonal anti-Rabbit Alexa Fluor 488 | Life Technologies | RRID:AB_143165 Cat# A-11008 | (1:200) |
| Antibody | Goat polyclonal anti-Rabbit Alexa Fluor 568 | Life Technologies | RRID:AB_10563566 Cat# A-11036 | (1:200) |
| Commercial assay or kit | Alexa Fluor 488 Tyramide SuperBoost Kit, goat anti-mouse IgG | ThermoFisher Scientific | Cat# B40912 | (1:200) |
| Commercial assay or kit | RNAscope Multiplex Fluorescent Kit v2 | Advanced Cell Diagnostics | Cat# 323110 | |
| Commercial assay or kit | RNAscope 2.5 HD Assay – RED | Advanced Cell Diagnostics | Cat# 322350 | |
| Commercial assay or kit | TSA Plus Cyanine 3 System | Perkin Elmer | Cat# NEL744001KT | |
| Commercial assay or kit | TSA Plus Fluoresceine System | Perkin Elmer | Cat# NEL771B001KT | |

*Continued*

| Reagent type (species) or resource | Designation | Source or reference | Identifiers | Additional information |
|---|---|---|---|---|
| Commercial assay or kit | RNeasy Micro Kit | QIAGEN | Cat# 74004 | |
| Commercial assay or kit | DAB Peroxidase (HRP) Substrate Kit (With Nickel) | Vector Laboratories | RRID:AB_2336382 Cat# SK4100 | |
| Commercial assay or kit | Chromium Single Cell 30 GEM, Library and Gel Bead Kit v3 | 10x Genomics Inc | Cat#1000092 | |
| Software, algorithm | ImageJ | NIH | RRID:SCR_003070 | |
| Software, algorithm | Ingenuity Pathway Analysis | Qiagen.Inc | RRID:SCR_008653 | |
| Software, algorithm | GraphPad Prism | GraphPad Software | RRID:SCR_002798 | |
| Software, algorithm | Seurat | Satija lab | RRID:SCR_016341 | |
| Software, algorithm | Monocle3 | Trapnell lab | RRID:SCR_018685 | |
| Software, algorithm | Cell ranger | 10X Genomics.Inc | RRID:SCR_017344 | |
| Software, algorithm | BWA | PMID:19451168; PMID:20080505 | RRID:SCR_010910 | |
| Software, algorithm | MACS | PMID:18798982 | RRID:SCR_013291 | |

## Animals

The following mice were used in this study: *Osr2-Cre* (gift from Rulang Jiang, Cincinnati Children's Hospital; *Chen et al., 2009*), *Runx2* floxed mice (gift from Dr. Takeshi Takarada, Okayama University, Japan; *Takarada et al., 2013*), *ROSA26loxp-STOP-loxp-tdTomato* conditional reporter (JAX#007905, *Madisen et al., 2010*) and *Twist1* floxed (MMRRC_016842-UNC; *Bildsoe et al., 2009*). To generate *Osr2-Cre;Runx2^{fl/fl}* mice, we crossed *Osr2-Cre;Runx2^{fl/+}* mice with *Runx2^{fl/fl}* mice. To generate *Osr2-Cre;Runx2^{fl/fl};Twist1^{fl/+}* mice, we bred *Osr2-Cre;Runx2^{fl/+}* mice with *Runx2^{fl/fl};Twist1^{fl/+}* mice. To generate *Osr2-Cre;tdTomato^{fl/fl}* mice, we crossed *Osr2-Cre;tdTomato^{fl/+}* mice with *tdTomato^{fl/fl}* mice. All mice were genotyped as previously described. All mice were used for analysis without consideration of sex. All studies were performed with the approval of the Institutional Animal Care and Use Committee (IACUC) at the University of Southern California. All the animals were handled according to approved IACUC protocol #9320 of the University of Southern California.

## MicroCT analysis

All microCT scans were performed using a SCANCO μCT50 device at the University of Southern California Molecular Imaging Center. Samples were scanned with the X-ray source at 70 kVp and 114 μA, and the data were collected at a resolution of 10 μM. Morphometric analysis was performed using the AVIZO 7.1 software package. Three biological replicates were performed. Measurements of hard tissues are based on the landmarks defined previously (*Ho et al., 2015*).

## Histological examination

Samples were fixed in 10% formalin, then decalcified in 10% EDTA followed by ethanol dehydration and paraffin embedding. Serial sections of 7 μm thickness were used for morphological analysis. These sections were stained using Hematoxylin and Eosin (H and E) following standard methods. Sections were imaged on a Keyence BZ-X710 microscope.

### *In situ* RNAscope hybridization

Mouse embryos were collected at E14.5 or E15.5 and fixed in 10% formalin. Samples were dehydrated with 15% and then 30% sucrose and embedded in OCT compound (Sakura, Tissue-Tek, Cat. 4583). OCT-embedded samples were sectioned at 8 µm on a cryostat. RNAScope 2.5 HD assay – red (Advanced Cell Diagnostics, Newark, CA, 322360) and RNAScope multiplex fluorescent v2 assay (Advanced Cell Diagnostics, 323100) were used for *in situ* hybridization according to the manufacturer's instructions.

Probes from Advanced Cell Diagnostics for *Myod1* (316081), *Scx* (439981), *Twist1* (414701), *Aldh1a2* (447391), and *Hic1* (464131), *Tfap2b* (536371), *Tbx22* (426511), *Ibsp* (415501), *Col2a1* (407221), *Tbx15* (558761), *Top2a* (491221), and *tdTomato* (317041) were used in this study.

### Immunofluorescence staining

Sections were processed with antigen-retrieval buffer (Vector Labs, Burlingame, CA, H-3300) for 15 min at 100℃, followed by 1% triton (Sigma Aldrich, St. Louis, MO, T8787) treatment for 10 min at room temperature. Afterwards, sections were incubated with blocking reagent (PerkinElmer, Waltham, MA, FP1012) for 1 hr at room temperature and the primary antibody overnight at 4℃. Alexa-conjugated secondary antibodies were used to show the fluorescence signal at 1:200 dilution. For myoblast determination protein 1 (MyoD), poly HRP-labeled goat anti-mouse IgG (ThermoFisher Scientific, Waltham, MA, B40912) was used as a secondary antibody and Alexa Fluor 488/594 Tyramide SuperBoost kit (PerkinElmer, Waltham, MA,NEL771B001KT, NEL774001KT) were used to develop the signal. Sections were counterstained with DAPI and imaged using a Leica DMI 3000B.

The following antibodies were used for immunostaining: Runx2 (Cell signaling technology, Danvers, MA, 12556S; 1:100), MyoD (DAKO, Carpinteria, CA, M3512; 1:25), myosin heavy chain (MHC; DSHB, Iowa City, IA, P13538; 1:10), active caspase 3 (Casp3; Cell signaling technology, Danvers, MA, 9661S; 1:100), and BrdU (BrdU; Abcam, Cambridge, UK, ab6326; 1:100). Anti-mouse, anti-rat and anti-rabbit Alexa Fluor 488 and 568 were used as secondary antibodies (A-11001, A-11004, A-11006, A-11008, A-11036, Thermofisher Scientific, Waltham, MA, 1:200).

### Single-cell RNA sequencing

Soft palate tissue (posterior third of the palatal region) was digested from E13.5, E14.5, and E15.5 controls and E14.5 *Osr2-Cre;Runx2^{fl/fl}* embryos by TrypLE express enzyme (Thermo Fisher Scientific, Waltham, MA) at 37℃ with shaking at 600 rpm for 20 min. Single-cell suspension was prepared according to the 10X Genomics sample preparation protocol. Seventeen thousand cells were loaded into the 10X Chromium system and prepared for single-cell library construction using the 10X Genomics Chromium single cell 3' v3 reagent kit. Sequencing was performed on the Novaseq 6000 platform (Illumina, San Diego, CA). Library quality control, sequence alignment, and read counts were analyzed using the CellRanger pipeline version 3.0.2. Raw read counts from each single cell in each sample were analyzed using Seurat R package (*Stuart et al., 2019*). Cell clusters and variably expressed genes in each cluster were identified by using Log Normalize, Find Variable Genes, Scale Data, and RunPCA functions. Seurat three package was used to combine the single-cell data from three stages as well as E14.5 control and *Osr2-Cre;Runx2^{fl/fl}* embryos to perform the integration analysis. Shared variances between different datasets were identified using the function FindIntegrationAnchors, then Seurat objects were processed using IntegrateData function. Scaledata, PCA, and UMAP visualization were then used for downstream analysis and visualization. Pseudotime trajectory analysis was done by Monocle three using Seurat 3 UMAP embedding to show cell fate restriction of CNC-derived soft palate mesenchymal cells across three development stages. Gene ontology and pathway analysis of enriched genes in different CNC-derived clusters was performed using Ingenuity Pathway Analysis (QIAGEN. Inc, Hilden, Germany).

### ATAC-seq

Single-cell suspension was prepared from the soft palate of E13.5-E14.0 control mice as described above and processed to generate ATAC-seq libraries according to a published protocol (*Buenrostro et al., 2015*). Sequencing was performed on the NextSeq 500 platform (Illumina, San Diego, CA). ATAC-seq reads were aligned to the UCSC mm10 reference genome using BWA-MEM (*Li, 2013*). ATAC-seq peaks were called by MACS2 (*Zhang et al., 2008*). Peaks were annotated and

known transcription factor binding motifs were analyzed in the ATAC-seq peaks by HOMER (*Heinz et al., 2010*).

## RNA sequencing

Soft palate tissue was collected from control and *Osr2-Cre;Runx2$^{fl/fl}$* embryos at E14.5. mRNA was isolated using RNeasy Micro Kit (QIAGEN, Hilden, Germany, 74404). Samples with RNA integrity number (RIN) >9.0 were used for cDNA library construction and sequencing by UCLA Technology Center for Genomics and Bioinformatics. Pair-end reads with 150 cycles sequencing were performed on Illumina NextSeq 500 platform. Sequence reads were trimmed and aligned using STAR (version 2.6.1d) using mm10 as the reference genome. Read counts were normalized using the upper quartile and differential expression was calculated using gene-specific analysis on the Partek Flow platform (Partek Inc, St. Louis, MO).

## Statistical analysis

T-tests were performed for statistical analysis using GraphPad Prism 7. Statistical data are presented as mean ± SEM.

## Acknowledgements

We thank Bridget Samuels and Linda Hattemer for critical reading of the manuscript. We gratefully acknowledge the USC Bioinformatics Service for providing computing resources and assistance with data analysis.

## Additional information

### Funding

| Funder | Grant reference number | Author |
| --- | --- | --- |
| NIH | R01 DE012711 | Yang Chai |
| NIH | U01 DE028729 | Yang Chai |

The funders had no role in study design, data collection and interpretation, or the decision to submit the work for publication.

### Author contributions

Xia Han, Conceptualization, Data curation, Software, Formal analysis, Validation, Investigation, Methodology, Writing - original draft; Jifan Feng, Investigation, Methodology; Tingwei Guo, Thach-Vu Ho, Courtney Kyeong Cho, Eva Janeckova, Jinzhi He, Fei Pei, Jing Bi, Brian Song, Validation; Yong-Hwee Eddie Loh, Yuan Yuan, Software; Jingyuan Li, Junjun Jing, Investigation; Yang Chai, Conceptualization, Resources, Formal analysis, Supervision, Funding acquisition, Investigation, Writing - review and editing

### Author ORCIDs

Jifan Feng (iD) https://orcid.org/0000-0002-9944-2604
Yuan Yuan (iD) https://orcid.org/0000-0002-7742-9433
Thach-Vu Ho (iD) http://orcid.org/0000-0001-6293-4739
Junjun Jing (iD) http://orcid.org/0000-0001-5745-5207
Yang Chai (iD) https://orcid.org/0000-0003-2477-7247

### Ethics

Animal experimentation: All studies were performed with the approval of the Institutional Animal Care and Use Committee (IACUC) at the University of Southern California. All the animals were handled according to approved IACUC protocol #9320 of the University of Southern California.

Decision letter and Author response
Decision letter https://doi.org/10.7554/eLife.62387.sa1
Author response https://doi.org/10.7554/eLife.62387.sa2

## Additional files

### Supplementary files

• Transparent reporting form

### Data availability

Sequencing data have been deposited in GEO under accession codes GSE155928.

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
