## [Decision Letter]

**Acceptance summary:**

This study reports a systematic single-cell RNA-seq analysis of soft palate development that identified key roles for the RUNX2 and TWIST1 transcription factors in regulating soft palate morphogenesis. Whereas *Runx2* is a known master regulator of osteogenesis, experiments here demonstrate *Runx2* function is required for perimysial cell differentiation during soft palate myogenesis. Furthermore expression of *Twist1* was significantly increased in the perimysial mesenchyme cells and that genetically inactivating one *Twist1* allele was able to rescue soft palate developmental defects in *the Osr2-KICre;Runx2^fl/fl^* mice. These findings provide significant new insight into the mechanisms of soft palate development, a clinically important but currently understudied area of craniofacial biology.

**Decision letter after peer review:**

Thank you for submitting your article "*Runx2-Twist1* interaction coordinates cranial neural crest guidance of soft palate myogenesis" for consideration by *eLife*. Your article has been reviewed by three peer reviewers, and the evaluation has been overseen by a Reviewing Editor and Kathryn Cheah as the Senior Editor. The following individual involved in review of your submission has agreed to reveal their identity: Kyoko Oka (Reviewer #3).

The reviewers have discussed the reviews with one another and the Reviewing Editor has drafted this decision to help you prepare a revised submission.

Summary:

This manuscript reports a systematic single-cell RNA-seq analysis of soft palate development and identified key roles for the RUNX2 and TWIST1 transcription factors in regulating soft palate morphogenesis. Whereas *Runx2* is a known master regulator of osteogenesis, this manuscript demonstrates that *Runx2* function is required for perimysial cell differentiation during soft palate myogenesis. Further data showed that expression of *Twist1* was significantly increased in the perimysial mesenchyme cells and that genetically inactivating one *Twist1* allele was able to rescue soft palate developmental defects in the *Osr2-KICre;Runx2^fl/fl^* mice. The data were well organized and the manuscript well written. The results and conclusions provide significant new insight into the mechanisms of soft palate development, a clinically important but currently understudied area of craniofacial biology. The sc-RNA-seq data sets should be a significant resource to the community

Essential revisions:

1) The main drawback identified by the reviewers is the *Runx2/Twist1* mechanism proposed, which the reviewers feel is problematic based on the presented data. In particular the use of the ATAC-seq data. The known mode of *Twist1/Runx2* interactions is that *Twist1* regulates *Runx2* activity by binding to it, thus preventing *Runx2* from binding DNA. However, this point is immediately confused by suggesting that both bind to DNA upstream of the *Aldh1a2* promoter through the use of ATAC-seq. Importantly, the finding that there are putative *Twist1* and *Runx2* binding sites in this region is not validated. The issue is further complicated the binding sites for *Twist1* and *Runx2* are over 20 kb apart, suggesting they do not identify a single enhancer element. Moreover there is no functional data provided to show that either of these regions are functional enhancers. The activities of *Twist1* and *Runx2* need to be carefully spelled out. The observations were found to be confusing – for example removal of *Runx2* leads to repression of *Aldh1a2* expression while *Twisty* expression increases. The finding that removing one copy of *Twist1* rescues *Aldh1a2* expression in the LVP does not prove that *Runx2* positively regulates *Aldh1a1* expression. An alternative explanation is that *Runx2* normally functions by repressing *Twist1*. In revising the paper the authors need to clearly state how they think *Twist1* and *Runx2* are working while also providing alternative explanations. This does not require additional work but does require deeper consideration and more thoughtful writing. Also, a model of *Twist1/Runx2* interaction (a flow diagram, not one incorporating tissue level signaling) would be helpful.

2) Please pay attention to improving the presentation of data in which colors appear over- or under-saturated. For example it appears that *Twist1* expression is almost gone when only losing one copy. Other figures bring up similar questions concerning areas of staining that are ignored or not well-described, while other areas are richly described. It is also noted that none of the images use confocal microscopy, though determination of co-expression is made throughout the paper. Many of these concerns could be addressed without additional data. The lack of confocal sections would be acceptable but needs to be acknowledged.

3) Since *Runx2* is a master regulator of osteogenesis and the *Osr2KI-Cre;Runx2^fl/fl^* mutant shown in Figure 3B appears to have a defect in the palatal process of the palatine (with a large gap in the middle of the posterior region of the hard palate bone), images of skeletal preparations should be included in this manuscript with description of the relevant palatal and pterygoid process defects. In addition, it needs to be discussed how the posterior hard palate bone defects could secondarily disrupt the palatine aponeurosis and integration/orientation of the soft palate muscles. If the palatal bone defects were variable in the *Osr2KI-Cre;Runx2^fl/fl^* mice, whether the skeletal defects correlated with the variability of soft palate cleft should also be noted. Related to this point, it will be of great value to also document whether the *Osr2KI-Cre;Runx2^fl/fl^;Twist1^fl/+^* mice exhibited rescue of the palatal and pterygoid bone defects and discuss how that could be relevant to the rescue of the soft palate.

---

## [Author Response]

Essential revisions:1) The main drawback identified by the reviewers is the Runx2/Twist1 mechanism proposed, which the reviewers feel is problematic based on the presented data. In particular the use of the ATAC-seq data. The known mode of Twist1/Runx2 interactions is that Twist1 regulates Runx2 activity by binding to it, thus preventing Runx2 from binding DNA. However, this point is immediately confused by suggesting that both bind to DNA upstream of the Aldh1a2 promoter through the use of ATAC-seq. Importantly, the finding that there are putative Twist1 and Runx2 binding sites in this region is not validated. The issue is further complicated the binding sites for Twist1 and Runx2 are over 20 kb apart, suggesting they do not identify a single enhancer element. Moreover there is no functional data provided to show that either of these regions are functional enhancers. The activities of Twist1 and Runx2 need to be carefully spelled out. The observations were found to be confusing – for example removal of Runx2 leads to repression of Aldh1a2 expression while Twist1 expression increases. The finding that removing one copy of Twist1 rescues Aldh1a2 expression in the LVP does not prove that Runx2 positively regulates Aldh1a1 expression. An alternative explanation is that Runx2 normally functions by repressing Twist1. In revising the paper the authors need to clearly state how they think Twist1 and Runx2 are working while also providing alternative explanations. This does not require additional work but does require deeper consideration and more thoughtful writing. Also, a model of Twist1/Runx2 interaction (a flow diagram, not one incorporating tissue level signaling) would be helpful.

We agree with the reviewers. The long distance between *Runx2* and *Twist1* binding sites suggests they probably do not form a complex to antagonize each other’s functions on the same downstream target at the transcriptional level. We also agree with the reviewers that it is hard to say if downregulation of *Aldh1a2* is directly caused by loss of *Runx2* based on the current data. It is possible that the reduction of *Aldh1a2* in *Runx2* mutants and restoration of *Aldh1a2* in *Twist1* heterozygous rescue are both caused by repression of *Aldh1a2* by *Twist1*. To validate this, direct binding of *Twist1* at the putative sites of the *Aldh1a2* gene locus and analysis of *Twist1* binding and activity changes of enhancers need to be performed in the future.

The corresponding discussion and flow diagram of *Runx2/Twist1* interaction model (Figure 6—figure supplement 1) in the text has been modified to reflect these points.

2) Please pay attention to improving the presentation of data in which colors appear over- or under-saturated. For example it appears that Twist1 expression is almost gone when only losing one copy. Other figures bring up similar questions concerning areas of staining that are ignored or not well-described, while other areas are richly described. It is also noted that none of the images use confocal microscopy, though determination of co-expression is made throughout the paper. Many of these concerns could be addressed without additional data. The lack of confocal sections would be acceptable but needs to be acknowledged.

We thank the reviewer for the comments. We have repeated the relevant experiments and retaken the pictures.

3) Since Runx2 is a master regulator of osteogenesis and the Osr2KI-Cre;Runx2^fl/fl^ mutant shown in Figure 3B appears to have a defect in the palatal process of the palatine (with a large gap in the middle of the posterior region of the hard palate bone), images of skeletal preparations should be included in this manuscript with description of the relevant palatal and pterygoid process defects. In addition, it needs to be discussed how the posterior hard palate bone defects could secondarily disrupt the palatine aponeurosis and integration/orientation of the soft palate muscles. If the palatal bone defects were variable in the Osr2KI-Cre;Runx2^fl/fl^ mice, whether the skeletal defects correlated with the variability of soft palate cleft should also be noted. Related to this point, it will be of great value to also document whether the Osr2KI-Cre;Runx2^fl/fl^;Twist1^fl/+^ mice exhibited rescue of the palatal and pterygoid bone defects and discuss how that could be relevant to the rescue of the soft palate.

Thank you for these comments.

1) We have performed microCT analysis of both hard and soft tissues of *Runx2* mutants and control mice. We observed palatine bone and pterygoid process defects along with the soft tissue defects in the *Runx2* mutant mice (Figure 3—figure supplement 2A-J), and the severity of skeletal defects was associated with the variability of the soft palate cleft. In *Runx2* mutants with soft palate clefts, the palatine bones were missing, and the pterygoid plate was significantly smaller in length and height than in control mice (Figure 3—figure supplement 2A-D, I). In *Runx2* mutants without clefts, although the length, width and height of the palatine bone and the length of pterygoid plate appeared smaller than those of control mice, those differences were not statistically significant. However, the height of the pterygoid plate in *Runx2* mutants without clefts was significantly shorter than in control mice (Figure 3—figure supplement 2E-H, J). Hence, only the pterygoid plate defects were consistent for all the *Runx2* mutants.

2) We analyzed the aponeurosis by performing RNAscope *in situ* analysis of *Scx* in the soft palate regions of both control and *Runx2* mutant mice at P0 (Figure 3—figure supplement 4). In *Runx2* mutant mice, the aponeurosis was thinner, and it did not stretch from the lateral-oral to the medial-nasal direction as it did in control mice. This probably causes the abnormal attachment of the TVP. However, for the LVP, we observed the muscle defects as early as E14.5 (Figure 4A-B), when palatine bones start to condense and differentiate, and it is before the palatal shelves’ fusion in the LVP region. This suggests that the defects of the LVP muscle are primary, independent of posterior hard palate bone and aponeurosis defects in *Runx2* mutant mice.

3) We collected *Runx2/Twist1* compound mutant rescue samples and performed hard tissue CT and histological analysis.

Since the palatine bone phenotypes in *Runx2* mutant mice varied, and the differences in those with less severe bone phenotypes were not significant (Figure 3—figure supplement 2J), it is difficult to conclude whether the palatine bone defects were rescued or not. We nevertheless collected *Runx2/Twist1* compound mutant rescue mice for hard tissue CT analysis (Author response image 1). Based on the CT analysis result, we observed that there were no significant differences between the palatine bones of *Runx2/Twist1* compound mutant rescue mice and *Runx2* mutants with less severe skeletal phenotypes. Moreover, as we mentioned earlier, the defects of the LVP muscle and the rescue of perimysial marker (*Aldh1a2*) expression appear as early as E14.5 (Figure 6J-L), when the palatine bones start to condense and differentiate. Taking these results together, it appears that the LVP muscle defect is unlikely as the result of defect in hard palate bone.

Interestingly, the height of the pterygoid plate was significantly affected in all the *Runx2* mutants, so we analyzed the pterygoid plate in control, *Runx2* mutants and *Runx2/Twist1* compound mutant rescue mice histologically and measured their heights. We observed partial rescue of pterygoid plate defects caused by loss of *Runx2* in *Runx2/Twist1* compound mutant mice (Figure 6B-H).

**Author response image 1. sa2fig1:** Deletion of *Runx2* in CNC-derived cells gives rise to hard tissue defects in *Osr2-Cre;Runx2^fl/fl^* mice. (A-C) Isolated palatine bones and sphenoid bones from control (A), *Osr2-Cre;Runx2^fl/fl^* (B) and *Osr2-Cre;Runx2^fl/fl^;Twist1^fl/+^* mice (C). Red dashed lines outline the pterygoid plate. Blue dashed lines outline the pterygoid plate. (D) Quantification of the size (length, width and height) of the palatine bone and of the pterygoid plate (length and height) from control (red bars), *Osr2-Cre;Runx2^fl/fl^* (blue bars) and *Osr2-Cre;Runx2^fl/fl^;Twist1^fl/+^* (yellow bars) mice (N=2). Scale bars in A-C: 0.6 mm.